# Application of a Bayesian Network Based on Multi-Source Information Fusion in the Fault Diagnosis of a Radar Receiver

**DOI:** 10.3390/s22176396

**Published:** 2022-08-25

**Authors:** Boya Liu, Xiaowen Bi, Lijuan Gu, Jie Wei, Baozhong Liu

**Affiliations:** 1Radar Faculty, Ordnance NCO Academy, Army Engineering University of PLA, Wuhan 430075, China; 2Hubei Key Laboratory of Intelligent Robot, Wuhan Institute of Technology, Wuhan 430073, China

**Keywords:** radar system, fault diagnosis, multi-source information fusion, Bayesian network, device-level

## Abstract

A radar is an important part of an air defense and combat system. It is of great significance to military defense to improve the effectiveness of radar state monitoring and the accuracy of fault diagnosis during operation. However, the complexity of radar equipment’s structure and the uncertainty of the operating environment greatly increase the difficulty of fault diagnosis in real life situations. Therefore, a Bayesian network diagnosis method based on multi-source information fusion technology is proposed to solve the fault diagnosis problems caused by uncertain factors such as the high integration and complexity of the system during the process of fault diagnosis. Taking a fault of a radar receiver as an example, we study 2 typical fault phenomena and 21 fault points. After acquiring and processing multi-source information, establishing a Bayesian network model, determining conditional probability tables (CPTs), and finally outputting the diagnosis results. The results are convincing and consistent with reality, which verifies the effectiveness of this method for fault diagnosis in radar receivers. It realizes device-level fault diagnosis, which shortens the maintenance time for radars and improves the reliability and maintainability of radars. Our results have significance as a guide for judging the fault location of radars and predicting the vulnerable components of radars.

## 1. Introduction

With the continuous progress of modern science and technology, an endless variety of new weapons have changed modern combat [1]. In modern war, air attacks are often used as the main means of warfare. Now more than ever, gaining control of the air has become vitally important. Radars have thus become indispensable equipment in the air defense system of various countries.

A radar is an electronic device that can work all day and in all weather conditions [2]. Due to the demanding tasks and long operation time of radars, the equipment failure rate has increased markedly. A radar receiver is an important component of a radar system. It performs pre-selection, amplification, frequency conversion, filtering, demodulation, and digital processing on the echo signal received by the radar antenna while suppressing external interference clutter and internal noise so that the echo signal can maintain the target information as much as possible in order to conduct further specialized signal processing [3]. Once the radar receiver fails, the radar is not able to accurately detect the target, which affects the normal operation of the entire air defense system [4]. Therefore, it is of great significance to strengthen the condition monitoring of radar receivers, analyze and evaluate important parameters, diagnose faults in time, and quickly formulate solutions.

With the development of artificial intelligence technology and the improvement of signal processing accuracy, technologies such as wavelet transform [5,6], support vector machines [7,8], principal component analyses [9,10], artificial neural networks [11,12,13], and deep learning [14,15,16] have been widely used in the field of fault diagnosis. These methods are able to learn and mine historical data under certain functional constraints to discern the corresponding relationship of a data model and then approach the mapping mechanism implied in the system data to carry out fault detection and diagnosis [17]. Based on monitoring data from different sources and types, this method solves the problem of incomplete fault representation and has real-time capabilities. However, this method invariably depends on the accuracy of the mathematical model and the real-time nature of the data and is restricted by factors such as the number of data and the calculation efficiency of the model. In most cases, these models they are unable to explain their reasoning processes and results, and there are different degrees of defects in the treatment of uncertainty problems.

Bayesian networks, proposed by Judea pearl in 1988, have powerful uncertainty problem processing abilities and are widely used in computer intelligence science, industrial control, medical diagnosis, and other fields. Moreover, Bayesian network can effectively express and fuse multi-source information and have great advantages when identifying the faults caused by uncertainty and the correlation of complex equipment [18].

Traditional radar fault diagnosis is realized by built-in test equipment (BITE). BITE is distributed in each function module of the radar. It can detect faults in equipment components and identify the faulty components. The maintenance personnel can then replace the faulty parts to eliminate the fault. The diagnostic process of BITE is shown in Figure 1.

When using BITE technology for maintenance support, the average fault recovery time (MTBR) of a radar is less than 10 min, but this equipment comes with many shortcomings. BITE lacks the ability to detect and isolate intermittent faults. Moreover, due to the direct replacement of the faulty modules, the service life of the components is generally reduced, resulting in an increase of 10–20% in the manufacturing and maintenance cost of a radar [19]. How to accurately and quickly locate the fault point inside the component and build a circuit board-level or even device-level fault diagnosis system is an unsolved problem that is worthy of study.

However, due to the compact and complex structure of modern radars and the high internal integration of their components [20], if a bite circuit is embedded in the components, it will not only have great requirements in terms of space utilization but may also have a certain impact on the signals passing through the components. The echo signal reflected from the target received by a radar antenna is very weak, unstable, and cluttered [21]. The impact of bit circuit embedding on the echo signal cannot be ignored, which means that BITE embedded in the receiver component is not feasible.

In addition, there is no differentiation between fault phenomena and causes in the system operation. A fault often manifests itself in a variety of fault phenomena, and sometimes several faults will be reflected by only a single fault phenomenon. Therefore, the uncertainty between the fault phenomenon and the fault cause make fault diagnosis more complex [22].

In order to solve these problems, this paper proposes a Bayesian network fault diagnosis method based on multi-source information fusion technology. This method realizes intelligent fault detection and diagnosis in radar receiver and improves the service life of radar components. The contributions of this paper are summarized as follows:A variety of monitoring sensors are designed in the receiver. Based on multi-source information fusion technology, the prior diagnosis database and the real-time monitoring database reflected by the sensor are analyzed and fused to achieve the diversification, quantification, and standardization of the data.A Bayesian network fault diagnosis model is proposed and applied to the fault diagnosis analysis of a radar receiver. The analysis results show that this method is effective.In contrast to traditional BITE technology, our method can realize accurate device-level fault location and fault cause analysis and avoid the replacement of whole components, reducing the maintenance costs of radars.

The remainder of this paper is organized as follows. Section 2 introduces the fault diagnosis system of a radar receiver. Section 3 provides the sample data and calculation results. Section 4 presents an in-depth discussion of the calculation results. Finally, conclusions are drawn in Section 5.

## 2. Fault Diagnosis System of Radar Receiver

### 2.1. Radar Receiver

The input signal of the receiver is always very weak and needs to be amplified and filtered to meet professional signal processing requirements. Early radar receivers used multistage, high-frequency amplifiers to amplify the receive echo signals, which were called high-frequency amplification receivers [23]. Since then, superheterodyne receivers have become more widely used, which mainly rely on the fixed frequency intermediate frequency (IF) amplifier to amplify the signal. The basic structure of a superheterodyne receiver is shown in Figure 2.

A superheterodyne receiver is generally composed of a limiter, a radio frequency (RF) low noise amplifier (LNA), a sensitivity time control (STC) circuit, a RF filter, a mixer, an IF amplifier, an IF filter, a detection circuit, and a low-frequency power amplifier [24]. The echo signal received by the antenna enters the receiver through the transmitter/receiver (T/R) switch. The echo signal inherits the RF characteristics of the transmitted signal. The power of an echo signal is weak, but its frequency is high [25]. Therefore, the power of an echo signal needs to be amplified to meet the normal working power of the detector. However, the frequency of the echo signal needs to be down-converted into an IF signal. In practical radar receiver applications, especially when the working band is high and the bandwidth is wide, a secondary frequency conversion scheme is usually adopted. Once an IF signal that meets the requirements is obtained, the in-phase digital signal and quadrature digital signal are output through gain control and phase detection and sent to the digital signal processor [26]. The two local oscillator signals required for the mixer and the phase reference signal that are required for phase detection are generated by the frequency synthesizer.

### 2.2. Multi-Source Information Fusion Technology

The traditional fault diagnosis method only analyzes one or a small number of kinds of information from the machine state to extract information about the machine behavior. Practice has proven that although using one kind of information can sometimes identify faults in mechanical equipment, the diagnosis results obtained in many cases are not reliable. Due to the development of computers, signal processing, artificial intelligence, pattern recognition, and other technologies, a new fault diagnosis method based on multi-source information fusion technology has emerged [27].

Information fusion technology can be described as the automatic detection, association, estimation, and combination of multi-source data to obtain more accurate and reliable information or inferences. It is a global method of multi-dimensional data processing [28]. The fusion framework of multi-source information fusion technology is generally divided into three stages: data-level fusion, feature-level fusion, and decision-level fusion [29].

Data-level fusion is the direct fusion of the original data, which is mainly used for the fusion of data with relatively consistent information types, such as image analysis, signal processing, text data format conversion and so on. Data-level fusion maximizes the authenticity of the information, but its fault tolerance is weak. Additionally, the effect is not ideal for the fusion of information that includes many data types and complex relationships [30]. Feature-level fusion is used to conduct feature extraction from the information source and then to carry out association fusion on this basis. Feature-level fusion realizes the structural consistency of different types of information through feature extraction, retains ample information, provides support for later decision analysis, and improves real-time information processing ability, accuracy, and efficiency [31]. Decision-level fusion involves high-level fusion. Compared with data-level fusion and feature-level fusion, it has the greatest amount of information loss and the poorest accuracy, but it can process a wide range of data types with high flexibility, strong anti-interference ability, and good fault tolerance [32].

The multi-source information fusion fault diagnosis model used in our system is shown in Figure 3.

Data Source

The data sources of the system are divided into prior diagnosis data sources and real-time data sources. The prior diagnosis data are mainly derived from expert experience, a case library, and historical monitoring data. The key points and conclusions of inspections, maintenance means, and maintenance steps summarized by professional maintenance technicians are used to compile the expert experience and case library, which are generally embodied in paper documents, electronic manuals, or oral experience inheritance. The historical detection data source is the operation data of the radar monitoring system or BITE since the radar has been operational. The BITE system of the radar is equipped with a single sensor, which is usually located inside the receiver in order to monitor the most important components in the receiver [33]. The operation data of the radar monitoring system or BITE only locate the fault at the level of the subsystem or the whole unit. The faults recorded in the expert experience and case library also have certain contingencies and cannot accurately reflect the current situation of the radar in real time [34]. If only a priori diagnosis data are used as the data source for the fault diagnostic model, it is unable to meet the requirements of fault diagnostic accuracy. Therefore, a variety of sensors are installed in the radar system to constantly monitor and inform the user of the status of the radar receiver.

An information processing method that uses multiple sensors to monitor the same specific target can overcome the uncertainty and limitations of a single sensor, obtain a consistent interpretation and description of the measured object, and realize the corresponding decision-making and estimation [35]. The type and number of sensors placed in the radar receiver can be determined according to the structure of the receiver and the parameter type of the monitored components.

Considering the complexity of the receiver’s high-frequency processing circuit and detection output circuit, as well as the variability of the environment, it can be added voltage sensors, current sensors, detection circuits, temperature sensors, humidity sensors, sound sensors, and cameras to monitor the voltage and current of the transmission path, the amplitude and phase of the signal, the sound of the internal action response in the components, and the temperature and humidity inside the receiver so as to achieve multi-directional and multi-dimension real-time monitoring and recording.

2.Information Fusion

The original data collected from the data sources are diverse, so it is necessary to standardize the multi-source information before fusion.

It must carry out data standardization and format conversion on the a priori diagnostic data first, then convert them into a unified data format, and finally screen out useful information for correlation and calibration on the premise of consistent data types [36]. The data collected by various sensors may be waveforms, voltage values, current values, images, and so on. The data must be preprocessed to extract and identify the required information before fusion. Then the redundant, complementary, and conflicting information of the multiple sensors in the system is associated and calibrated by following certain rules. Finally, the information of the two data sources is fused to obtain a consistent description of the real situation of the measured object. This method provides more meaningful and valuable information for decision-making.

3.Decision Discrimination

Effective information and decision theory are used to infer results [37]. Commonly used decision theories include the Kalman filter, the Dempster–Shafer evidence theory, expert knowledge systems, artificial neural networks, etc. Bayesian network reasoning is used in this paper. Bayesian networks have a strong ability to deal with uncertain problems. The construction of a Bayesian network consists of the determination of the network nodes, the directed correlation between the nodes, and the posterior probability. After reasoning through a Bayesian network, the final result of the fault diagnosis is output.

### 2.3. Bayesian Network

#### 2.3.1. Brief Description

A Bayesian network is a directed graphic description based on a network structure that is the product of the integration of probability theory, graph theory, and decision theory [38].

It uses a structural directed graph to express the association relationship between the information elements and the influence node variables, uses the directed edge between nodes to connect the association relationship between elements, and uses conditional probability to express the influence degree of each information element.

Let H=X, L denote a directed acyclic graph where $X=\left\{X_{1},X_{2},\cdots ,X_{n}\right\}$ denotes the set of random variables and L denotes the set of edges of the Bayesian network, then the joint probability of X can be expressed as:(1)$$P\left(X_{1},X_{2},\cdots ,X_{n}\right)=\prod_{i=1}^{n}P\left(X_{i}|Parents(X_{i}\right)=P\left(X_{n}|X_{n-1},\cdots ,X_{1}\right)P\left(X_{2}|X_{1}\right)\;PX_{1}$$

The basis of Bayesian network reasoning is the probability relationship between variables or events.

A Bayesian network can be regarded as the joint probability distribution of a group of random variables, from which reasoning and decision-making can be carried out [39]. All kinds of information related to fault diagnosis and maintenance decisions can be incorporated into a Bayesian network structure, which can handle it uniformly in the form of nodes and fuse different parts effectively according to the correlations between the information.

In addition, it can learn and reason under limited, incomplete, and uncertain information conditions and make reasonable and quantifiable decisions to maximize the decision efficiency [40].

#### 2.3.2. Determination of Bayesian Network

The main functions of the receiver are frequency conversion and phase detection [41]. There are many types of faults in these two functions. This paper takes the fault that the IF signal output obtained after the secondary mixing of the receiver is abnormal as an example to analyze the possible factors causing the fault.

Abnormal IF signal output indicates a variety of fault phenomena. The fault phenomena that may be detected include abnormal signal frequency, inconsistent multi-channel signal phase, inconsistent amplitude, low signal power, abnormal control voltage, high noise, high clutter, ineffective STC modulation, and so on. The corresponding fault points of these fault phenomena are shown in Figure 4.

The power supply is the energy source of the active devices in electronic circuits. The performance of the power supply directly affects the performance of electronic circuits. The power supply can be said to be the “heart” of electronic systems. 

The frequency mixer, amplitude and phase corrector, amplitude limiter, frequency synthesizer, RF STC, power amplifier, and filter in the receiver all require electric energy, as well as a variety of power supplies with different voltages and capacities, to work normally.

The power converter is a special device in the radar receiver. It has two functions: voltage transformation and power supply. The external power supply enters the receiver, is transformed into the voltage values required by each component of the receiver through the power converter, and is then sent to each component to supply it with power. One power converter can convert the power supply to several voltage values, and the fault types of the power converter are also diverse. If a failure occurs in one voltage transformer branch of the power converter, the devices in the receiver that require a power supply from the transformer branch will not work. Once the power converter has a functional fault, there is no normal power supply for any of the electronic devices inside the receiver, meaning that the receiver will be in “paralysis”, as is shown in Figure 5.

By combining Figure 4 and Figure 5, the nodes and directed edges of the Bayesian network can be preliminarily determined so as to determine the Bayesian network model, as is shown in Figure 6.

F_1_ and F_2_ are two root nodes, which represent “abnormal IF signal output” and “power converter failure”, respectively. There are 21 causes that may lead to the F_1_ phenomenon, which are represented by E_1_–E_21_. The F_2_ phenomenon will make all components inoperative, resulting in the failure of the IF signal output and thus indirectly leading to the F_1_ phenomenon. See Table 1 and Table 2 for evidence definition and relevant information.

## 3. Results

Using the historical monitoring data of the radar, the conditional probability tables (CPTs) of all nodes can be determined in the fault diagnostic network. The prior probabilities of fault nodes F_1_ and F_2_ are shown in Table 3.

The parent node of the E_1_, E_2_, E_3_, E_4_, E_6_, and E_7_ events is F_1_. The parent node of the E_5_, E_10_, E_11_, E_12_, E_13_, E_14_, E_15_, E_17_, E_18_, E_19_, and E_21_ events is F_2_. E_1_ has the child nodes E_8_, E_9_, E_10_, and E_13_. If any of the child nodes fails, the fault phenomenon represented by E_1_ is bound to occur.

The failure of E_10_ and E_13_ may be caused by F_2_ power failure. Similarly, if E_17_, E_18_, or E_19_ fails, E_16_ and E_4_ will occur, resulting in F_1_ failure. By analogy, the posterior probabilities of other events are also extracted according to sample data processing and calculation. The results are shown in Table 4.

With the calculations of the monitoring data samples and Bayesian network model, the CPTs can be denoted. Under different conditions of F_1_ and F_2_, the conditional probabilities of E_1_–E_21_ are different, and the posterior probabilities in the table can be transformed into a more intuitive chart form, as is shown in Figure 7.

## 4. Discussion

### 4.1. F_1_ and F_2_ Sub-Node Evidence

From the previous analysis, it can be seen that if the IF signal output is abnormal or the power converter fails, six fault phenomena will be directly caused; that is, the first layer child of nodes E_1_, E_2_, E_3_, E_4_, E_6_, and E_7_ events of F_1_ and F_2_. According to the CPTs, the relevant conditional probability distribution diagram can be obtained, which is shown in Figure 8.

In Figure 8, it can be seen that the event probability of node E_4_ (low signal power) is the highest under any condition of F_1_ and F_2_.

Let $P_{t}$ be the power of the echo signal, $P_{t}$ be the power of the transmit signal, G be the receiving antenna gain, λ be the transmitted electromagnetic wave length, σ be the radar cross section, and *R* be the distance from the target to the radar. The Radar Equation can be obtained using the following equation:(2)$$P_{r}=\frac{P_{t}G^{2}\sigma \lambda^{2}}{{\left(4\pi \right)}^{3}R^{4}}$$

According to the radar equation, $P_{r}$ is inversely proportional to $R^{4}$. The farther the target that is detected, the weaker the power of the echo signal [42]. This is because the emitted electromagnetic wave will decay rapidly in the transmission process. The electromagnetic wave emitted by the radar is a radio frequency signal with great power, but the signal energy reflected by the target received by the radar is extremely weak and may be only 10^−7^–10^−6^ V. This signal can be received by the signal processor only after it is amplified to at least tens of volts. Therefore, whether the echo signal power can be amplified to a strength that the signal processor can recognize is one of the key methods of testing the performance of the receiver. The all-weather/all-day characteristics of the radar have higher requirements for the stability of the receiver power amplifier. The power gap between the echo signal received by the antenna and the IF signal transmitted to the signal processor is 10^7^–10^8^ orders of magnitude, which requires power amplifiers. Consequently, the failure of power amplifier efficiency to meet the requirements is the most likely cause of an unsatisfactory output signal from the receiver, which is consistent with the result of the maximum probability of node E_4_ shown in Figure 8.

### 4.2. E_4_ Sub-Node Evidence

The direct fault cause of E_4_ (low signal power) evidence is E_16_ (power amplifier under power). According to a previous circuit analysis, general superheterodyne receivers have the tertiary power amplifier RF LNA for the 1st IF amplifier and 2nd IF amplifier. If any of the three amplifiers fails, E_4_ (low signal power) evidence will be observed. The power of the echo is weak, but it inherits the high frequency of the transmitted signal. The RF low-noise amplifier is the first station for echo signal amplification. It should not only be able to withstand high-frequency signals but also undertake power amplifier tasks, which are required to suppress the interference of noise to improve the signal-to-noise ratio. Compared with the 1st IF amplifier and 2nd IF amplifier, the RF LNA is the most vulnerable device in the receiver, and it has a higher probability of failure. E_4_ has four sub-nodes, E_16_–E_19_, which correspond to power amplifier under power, RF LNA failure, 1st IF amplifier failure, and 2nd IF amplifier failure events, respectively. Their conditional probability distributions are shown in Figure 9.

In Figure 9, E_16_ has the highest probability because it is the direct cause of the low power of the signal. E_17_–E_19_ are the child nodes of E_16._ Among them, E_17_ (RF LNA failure) has the highest probability, which is in line with the circuit analysis results.

### 4.3. E_6_ and E_7_ Sub-Node Evidence

In addition to E_4_ (low signal power), which is the most likely fault phenomenon, there may also be abnormal output frequency, inconsistent phase, inconsistent amplitude, abnormal control voltage, more clutter, ineffective STC modulation, etc. It can be seen from Figure 8 that, compared with other fault phenomena, the probabilities of E_6_ (more clutter) and E_7_ (ineffective STC modulation) are very low and can basically be ignored. It is indicated that the clutter suppression and STC control effect are the best and the possibility of failure is lowest in the receiver. The conditional probability distribution of an RF STC fault is shown in Figure 10, and the conditional probability distributions of the sub-nodes of E_6_ (more clutter) are shown in Figure 11.

If the IF signal output is abnormal and the power converter has no faults, the probability of E_5_ (RF STC failure) is the highest.

The direct cause of more clutter is E_20_ (insufficient filtering) in the receiver. The filtering effect is basically subject to the three filters in the receiver. Under the same conditions, the failure probability of these three filters is almost the same, and because the probability of E_6_ is very low, the failure probability of the filter can also be ignored.

### 4.4. E_2_ and E_3_ Sub-Node Evidence

Generally speaking, the probability of an inconsistent phase or amplitude of the IF signal output by the receiver is not high. The probability of these two phenomena will increase only when the power converter fails, and the electronic components cannot work normally. The inconsistent phase and amplitude may be caused by the poor effect of the amplitude and phase corrector and amplitude limiter. The conditional probability distribution of E_2_ and E_3_ sub-nodes is shown in Figure 12.

In most cases, the probability of limiter failure is higher than that of other kinds of failures. When the power converter fails, the probability of amplitude and phase corrector failure increases.

### 4.5. E_1_ Sub-Node Evidence

In addition to amplifying the echo signal, the main task of the receiver is to reduce the frequency of the echo signal and convert the RF signal into an IF signal through down-conversion [43]. The abnormal frequency of the final IF output signal is uncommon. If the frequency is abnormal, it means that the mixing effect is not ideal; mixers and local oscillator signals are the factors that influence the mixing effect. The local oscillator signals come from the frequency synthesizer. The conditional probability distributions of the sub-nodes of E_1_ (abnormal signal frequency) are shown in Figure 13.

In the mixing process of the receiver, if the local oscillator signals are abnormal, it will report a fault in the frequency synthesizer and will not enter the mixer without warning. Therefore, compared with the local oscillator signals, the failure probability of the mixer is higher. The mixer is also an electronic device and needs a power supply. In the case of a power converter failure, the failure probability of the mixer is higher.

## 5. Conclusions

This paper provides a Bayesian fault diagnosis method based on multi-source information fusion. This method determines the Bayesian network model by processing and fusing expert experience data, a case library, historical monitoring data, and data obtained from various sensors according to the relationship between fault phenomena and causes, as well as the circuit structure characteristics of the radar receiver. Then, by calculating the conditional probability of each node, the device that is most likely to cause the fault is output, which proves the effectiveness of this method. This fault diagnosis method optimizes the component level fault alarm (which is completed by BITE at the moment), locates the fault points at the device level, avoids the waste of resources caused by the replacement of a whole piece during radar maintenance, and improves the efficiency and cost ratio of the maintenance process. In the future, the method proposed in this paper will be applied to the status monitoring and fault diagnosis of radar subsystems, such as transmitters, antenna feeders, recording terminal, power supplies, and so on, and the stability and accuracy of this method will be further improved through data collection and analysis.

## Figures and Tables

**Figure 1 sensors-22-06396-f001:**
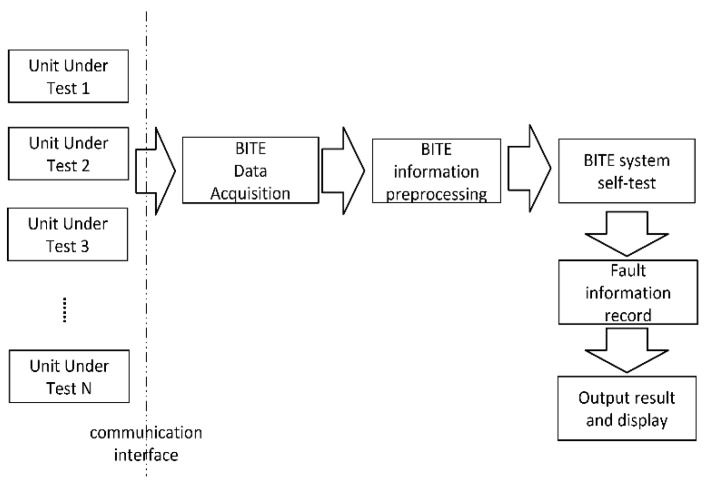
BITE diagnosis flow chat.

**Figure 2 sensors-22-06396-f002:**
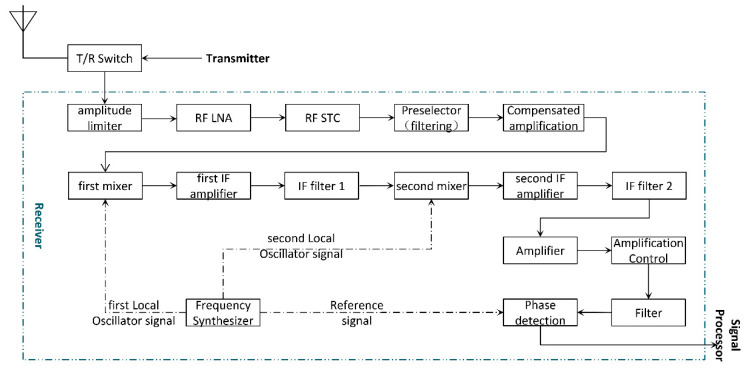
Structure Diagram of a Superheterodyne Receiver.

**Figure 3 sensors-22-06396-f003:**
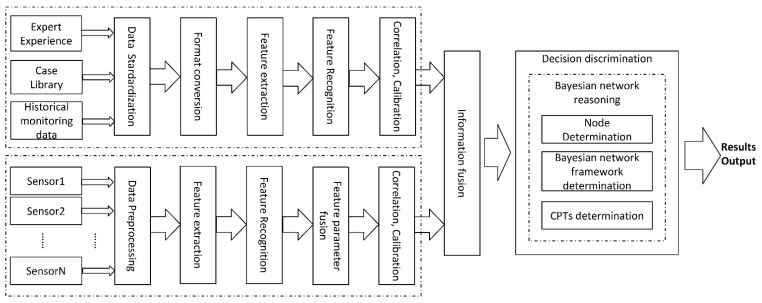
The multi-source information fusion fault diagnosis model.

**Figure 4 sensors-22-06396-f004:**
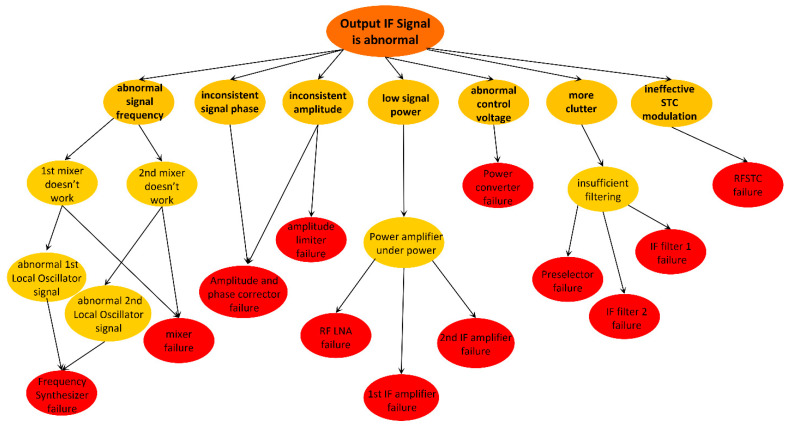
Relationship between abnormal IF signal fault phenomenon and fault points.

**Figure 5 sensors-22-06396-f005:**
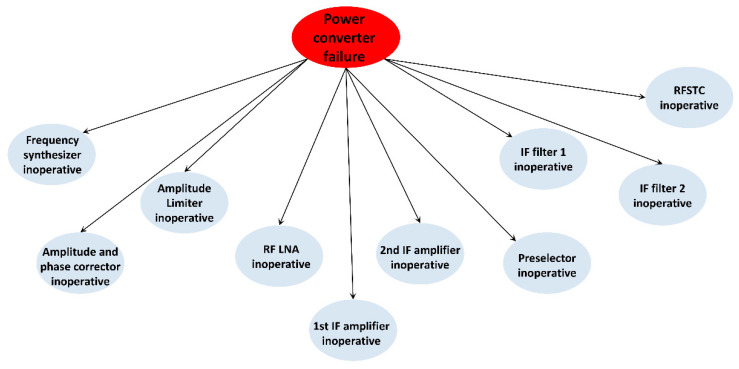
Relationship diagram of power converter fault.

**Figure 6 sensors-22-06396-f006:**
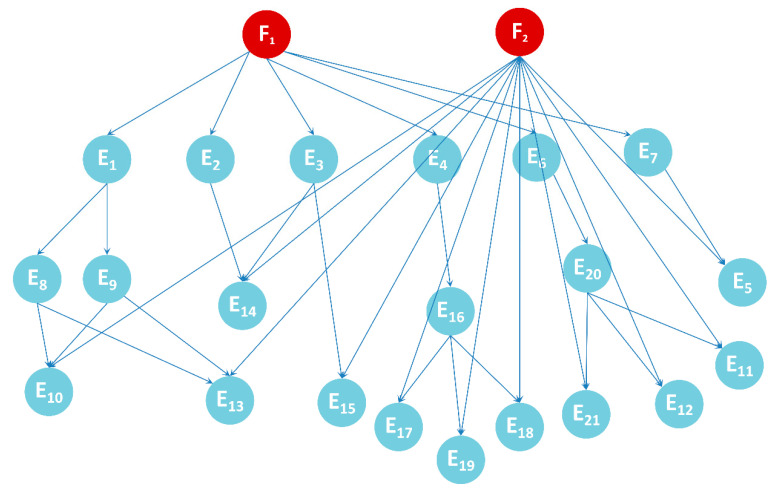
Bayesian network model.

**Figure 7 sensors-22-06396-f007:**
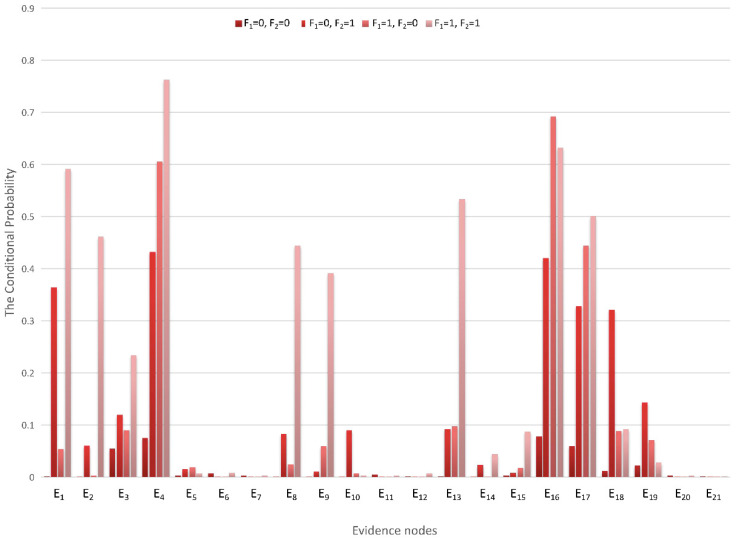
Conditional probability distribution diagram.

**Figure 8 sensors-22-06396-f008:**
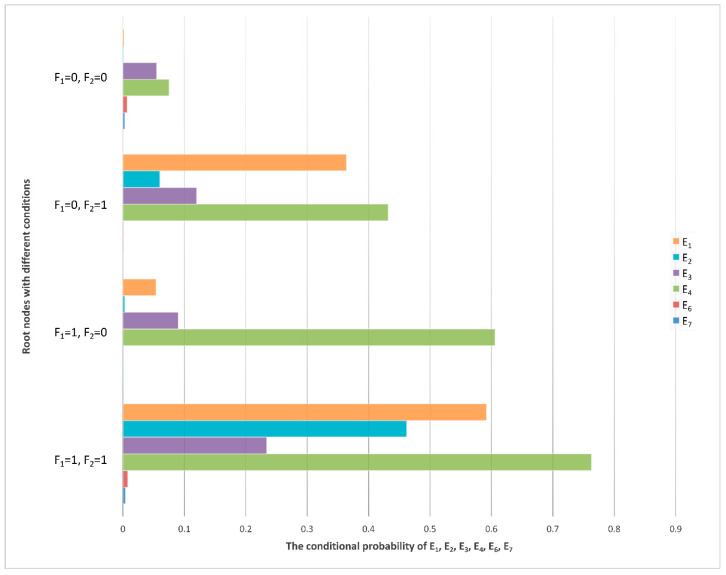
Conditional probability distribution diagram of the first layer sub-nodes.

**Figure 9 sensors-22-06396-f009:**
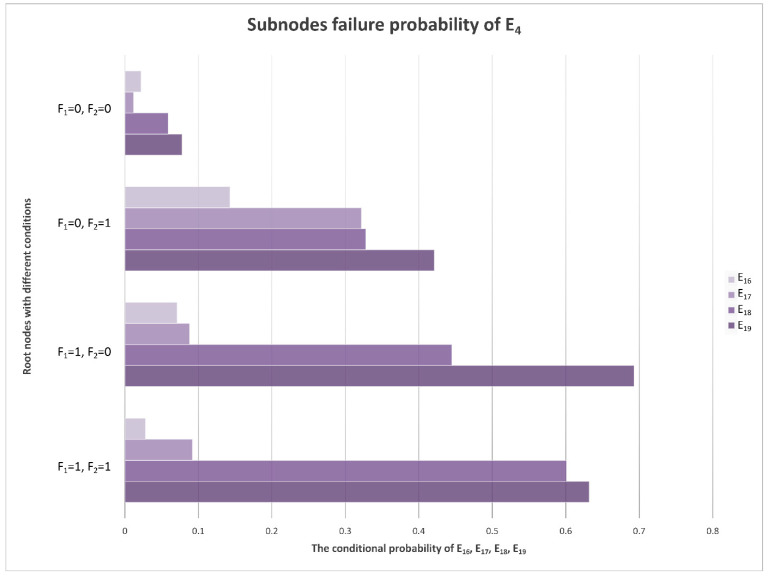
Conditional probability distribution diagram of E_4_’s sub-nodes.

**Figure 10 sensors-22-06396-f010:**
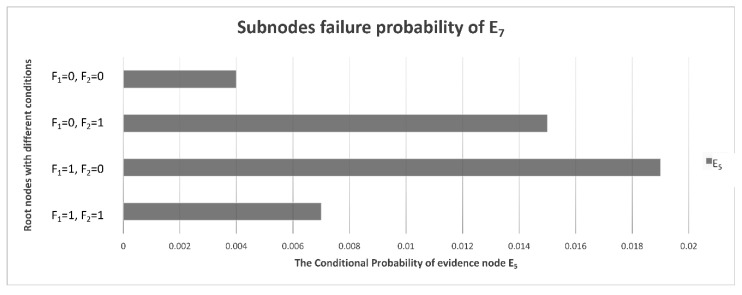
Conditional probability distribution diagram of E_5_’s sub-nodes.

**Figure 11 sensors-22-06396-f011:**
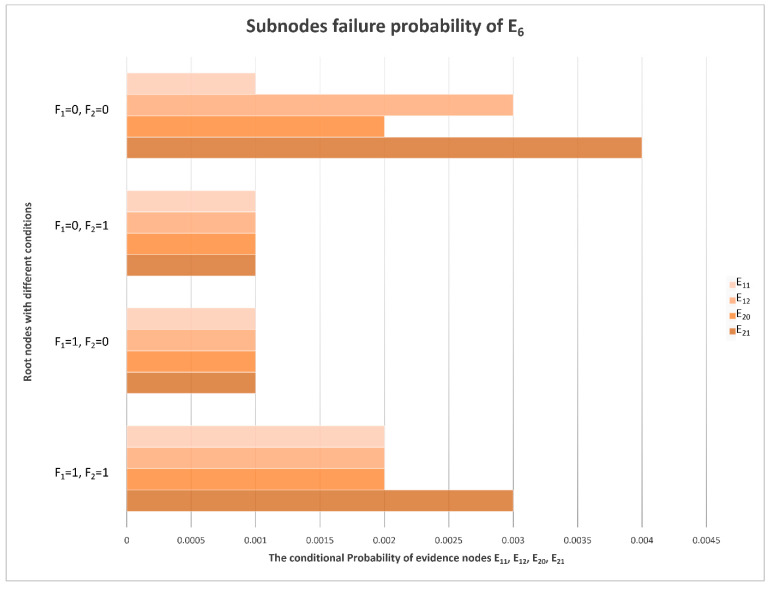
Conditional probability distribution diagram of E_6_’s sub-nodes.

**Figure 12 sensors-22-06396-f012:**
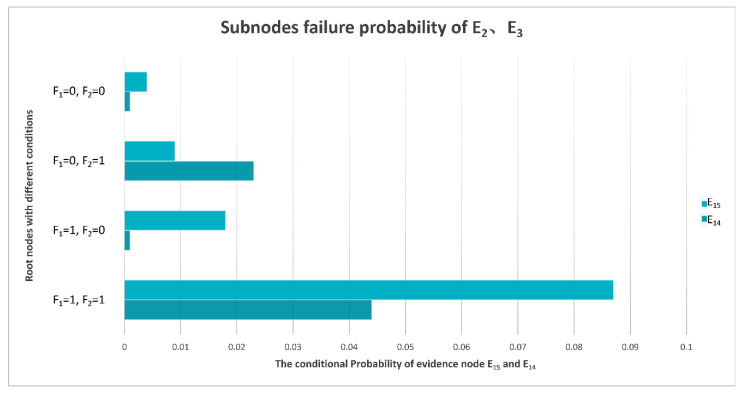
Conditional probability distribution diagram of sub-nodes of E_2_ and E_3_.

**Figure 13 sensors-22-06396-f013:**
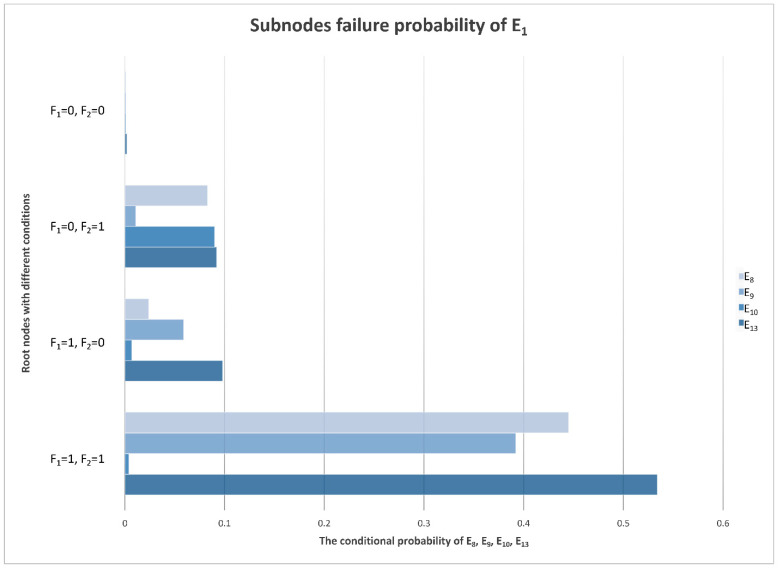
Conditional probability distribution diagram of E_1_’s sub-nodes.

**Table 1 sensors-22-06396-t001:** Description of faults.

Nodes	Node Description	Monitoring Parameters
F_1_	Output IF signal is abnormal	Fault indicator of Receiver
F_2_	Power converter failure	Voltage

**Table 2 sensors-22-06396-t002:** Description of evidence nodes.

Nodes	Node Description	Monitoring Parameters
E_1_	Abnormal signal frequency	Frequency of Echo signal
E_2_	Inconsistent signal phase	Phase of signal Voltage
E_3_	Inconsistent signal amplitude	Amplitude of signal
E_4_	Low signal power	Power of signal
E_5_	RF STC failure	Amplitude of signal
E_6_	More clutter	Spectrum of signal
E_7_	Ineffective STC modulation	Amplitude of signal
E_8_	1st mixer inoperative	Frequency of signal
E_9_	2nd mixer inoperative	Frequency of signal
E_10_	Frequency synthesizer failure	No signal
E_11_	2nd IF filter failure	Spectrum of signal
E_12_	1st IF filter failure	Spectrum of signal
E_13_	Mixer failure	Voltage
E_14_	Amplitude and phase corrector failure	Amplitude and phase of signal
E_15_	Amplitude limiter failure	Amplitude of signal
E_16_	Power amplifier under power	Power of signal
E_17_	RF LNA failure	Power of signal
E_18_	2nd IF amplifier failure	Power of signal
E_19_	1st IF amplifier failure	Power of signal
E_20_	Insufficient filtering	Spectrum of signal
E_21_	Preselector failure	Spectrum of signal

**Table 3 sensors-22-06396-t003:** The prior probabilities of nodes F_1_ and F_2_.

Nodes	State (True/False)	P(F_x_)
F_1_	1	P(F_1_) = 0.074
	0	P(F_1_) = 0.926
F_2_	1	P(F_2_) = 0.219
	0	P(F_2_) = 0.781

**Table 4 sensors-22-06396-t004:** The conditional probabilities of evidence nodes E_1_–E_21_.

Nodes	F_1_	F_2_	E_1_	E_2_	E_3_	E_4_	E_5_	E_6_	E_7_
Probability	0	0	0.002	0.001	0.055	0.075	0.004	0.007	0.003
	0	1	0.364	0.060	0.120	0.432	0.015	0.001	0.001
	1	0	0.054	0.003	0.090	0.606	0.019	0.001	0.001
	1	1	0.592	0.462	0.234	0.763	0.007	0.008	0.004
**Nodes**	**F_1_**	**F_2_**	**E_8_**	**E_9_**	**E_10_**	**E_11_**	**E_12_**	**E_13_**	**E_14_**
Probability	0	0	0.001	0.001	0.001	0.005	0.002	0.002	0.001
	0	1	0.083	0.211	0.090	0.001	0.001	0.092	0.023
	1	0	0.024	0.059	0.007	0.001	0.001	0.098	0.001
	1	1	0.445	0.392	0.004	0.003	0.007	0.534	0.044
**Nodes**	**F_1_**	**F_2_**	**E_15_**	**E_16_**	**E_17_**	**E_18_**	**E_19_**	**E_20_**	**E_21_**
Probability	0	0	0.004	0.078	0.059	0.012	0.022	0.004	0.002
	0	1	0.009	0.421	0.328	0.322	0.143	0.001	0.001
	1	0	0.018	0.693	0.445	0.088	0.071	0.001	0.001
	1	1	0.087	0.632	0.501	0.092	0.028	0.003	0.002

## Data Availability

Not applicable.

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
