# Peer review of "Application of a Bayesian Network Based on Multi-Source Information Fusion in the Fault Diagnosis of a Radar Receiver"

_sensors, 2022, doi:10.3390/s22176396_

Round 1
Reviewer 1 Report
This manuscript proposes a Bayesian network model to improve the effectiveness of radar receiver state monitoring and the accuracy of fault diagnosis in the process of operation. The idea is good; however, many issues should be addressed according to the following comments:
1) The overall presentation, readability, and more discussion analysis with new results are mandatory. Also, the manuscript suffers from a lot of language problems, please correct the language problems, it is weak from the Grammarly and sequences of events, I catch 19 errors by using a personal program. The paper must be proofread very carefully by a native speaker or a proofreading agent.
2) The "Abstract" section must be more intensively focused on the main idea directly and must contain the contribution of this manuscript with numerical result indicators.
3) The "Introduction" section should be made much more impressive and focused on the main idea directly by highlighting your contributions. The novelty of this manuscript must be explained simply and clearly in points at the end of the introduction section.
4) The authors must avoid the use of the pronoun "we" in the whole manuscript.
5) The introduction section should be enriched with up-to-date references by adding and citing the latest trends in fault diagnosis using machine learning and IoT technique. E.g., Effective IoT-based Deep Learning Platform for Online Fault Diagnosis of Power Transformers & Reliable Deep Learning and IoT-Based Monitoring System for Secure Computer Numerical Control Machines.
6) Check carefully all the abbreviation definitions, symbols, and standard units in the whole manuscript. I catch some errors and the other symbols are not defined, please cure them.
7) Most of the figures like a photo, the resolution and quality of result figures should be presented as close to the camera-ready format. Also, please don't use the symbol abbreviations on X-Y-axes, they must have the full name with their SI units.
8) The conclusion section should be rearranged, and numerical results should be pointed out. Also, the authors may propose some interesting problems as future work at the end of the conclusion.
Reviewer 2 Report
Dear Authors,
You have considered an interesting research subject. However, below are some notes you need to consider to improve the article.
11- The obtained results need to be better reflected by values or percentages in the abstract.
22- Also, the abstract is generally a little foggy, so please rewrite it to clarify it.
33- The paper contributions are not sufficiently clarified in the introduction section; thus, it is highly suggested that the authors clearly define their contributions.
44- Please write a short paragraph at the end of the introduction section to show how the paper is organized.
55- This sentence, “If only 234 one voltage transformer branch occurred failure, the devices in the receiver that need the 235 transformer power supply will not work” is unclear; please clarify.
6- What are the other artificial intelligence algorithms that could be applied in this work besides the Bayesian network algorithm?
7- In the conclusion section, please refer to the future prospects of this work.
88- There are some typos and grammatical errors in the manuscript; thus, it is strongly recommended that the whole work be proofread carefully.
Round 2
Reviewer 1 Report
All of my concerns are adjusted.
Reviewer 2 Report
Thanks for the accurate consideration of the given comments.